# Mechanical and Viscoelastic Properties of Carbon Fibre Epoxy Composites with Interleaved Graphite Nanoplatelet Layer

Barbara Palmieri [1], Ciro Siviello [2], Angelo Petriccione [3], Manuela Espresso [3], Michele Giordano [1], Alfonso Martone [1,*] and Fabrizia Cilento [1]

1   Institute of Polymers, Composite and Biomaterials, Italian National Research Council (CNR), 80055 Portici, NA, Italy; barbara.palmieri@ipcb.cnr.it (B.P.); michele.giordano@cnr.it (M.G.); fabrizia.cilento@ipcb.cnr.it (F.C.)
2   Jaber Innovation, 81030 Teverola, CE, Italy
3   Advanced Tools and Moulds srl, Zona Industriale ASI Stabilimento 13, 81030 Gricignano d'Aversa, CE, Italy; angelo.petriccione@atmsrl.com (A.P.); manuela.espresso@atmsrl.com (M.E.)
*   Correspondence: alfonso.martone@cnr.it

**Abstract:** The use of interleaving material with viscoelastic properties is one of the most effective solutions to improve the damping capacity of carbon fibre-reinforced polymer (CFRP) laminates. Improving composite damping without threatening mechanical performance is challenging and the use of nanomaterials should lead to the target. In this paper, the effect of a nanostructured interlayer based on graphite nanoplatelets (GNPs) on the damping capacity and fracture toughness of CFRP laminates has been investigated. High-content GNP/epoxy (70 wt/30 wt) coating was sprayed on the surface of CF/epoxy prepregs at two different contents (10 and 40 g/m$^2$) and incorporated at the middle plane of a CFRP laminate. The effect of the GNP areal weights on the viscoelastic and mechanical behaviour of the laminates is investigated. Coupons with low GNP content showed a 25% increase in damping capacity with a trivial reduction in the storage modulus. Moreover, a reduction in interlaminar shear strength (ILSS) and fracture toughness (both mode I and mode II) was observed. The GNP alignment and degree of compaction reached during the process were found to be key parameters on material performances. By increasing the GNP content and compaction, a mitigation on the fracture drop was achieved (−15%).

**Keywords:** damping; nanomaterials; carbon fibre laminates; interleaved layer





## 1. Introduction

Due to the trend towards high-speed, lightweight, automated, and multifunctional aerospace vehicles, the problems of induced vibration and noise are becoming increasingly relevant. Carbon fibre-reinforced polymer (CFRP) composites are commonly used in weight-sensitive structural applications concerning standard metallic structures due to their high stiffness-to-weight ratio [1]. Composites must satisfy the high requirement for vibration and noise reduction in the case of aeronautical vehicles, but also must be damage resistant and damage tolerant. Thus, many efforts have been made by worldwide researchers to simultaneously improve the fracture toughness and dissipation capacity of thermoset matrix composites without significantly adding weight or reducing in-plane mechanical properties.

Different strategies to improve the passive damping of composites include the use of hybrid fibres [2,3] or a high viscoelastic polymeric matrix [4,5] as interleaving damping materials, such as viscoelastic materials [6,7]. The last approach is the most promising since it improves the inherent capacity of laminates to dissipate vibrational energy, by the shearing motion of the viscoelastic layer [8,9]. Typically, interlaminar stresses arise at lamina interfaces in composite laminates, dissipating part of the total energy through interlaminar damping.

However, the addition of an interlayer usually deteriorates the elastic properties of the material [10,11]. Improving the damping and the interlaminar fracture toughness of composite materials and maintaining high stiffness and strength is still challenging.

Recently, nanomaterials have been employed to meet this requirement. It is well known that the use of nanomaterials can effectively increase the mechanical performances of polymers both in terms of elastic modulus and damping, thanks to the energy dissipation that occurs at the interface with the matrix [12]. In addition, 1D/2D nanomaterials can simultaneously improve composite damping and mechanical properties in terms of the elastic modulus and damping, thanks to the energy dissipation properties [13]. Specifically, it has been found that nanoparticles with a lamellar (2D) structure, such as graphene and its analogues (graphene oxide, GO, graphene nanoplatelets, GNP, etc.), can significantly improve both the damping capabilities of nanocomposites [14–16] and the fracture toughness [17,18]. Graphite nanoplatelets (GNPs) are known to have exceptional mechanical properties, i.e., high stiffness, strength, and toughness, which make them attractive candidates for reinforcement in composite materials [19–22]. Due to the 2D nature and high specific area of graphene, a significant increase in mode-I fracture toughness of polymers was found at extremely low loadings of nanoplatelets, thanks to the strong interfacial bonds and improved load transfer and crack resistance [23]. Composites with integrated damping features can be designed by appropriately choosing nanofillers, which are capable of improving the passive dissipation performance of laminates [24].

However, although nanofillers could improve the toughening mechanism of polymers when added to FRPs, the enhancement in fracture toughness is uncertain [25]. Ahmadi-Moghadam et al. [26] found that GNPs effectively and efficiently enhance the mode-I fracture toughness of the epoxy resin, but not the mode-II fracture toughness, due to nanoparticle/matrix debonding and the absence of filler bridging. An increase in both $G_{IC}$ and $G_{IIC}$ was found by Quan et al. [27] in the case of CFRP interleaved with MWCNT- doped polyphenylene–sulfide (PPS) veils with a 0.5 g/m$^2$ density, while a reduction of $-11\%$ $G_{IC}$ in the case of the GNPs/PPS veil of the same density due to agglomeration inhibited PPS fibre/epoxy adhesion. On the same path, Nagi et al. [28] investigated the effect of mode-I and mode-II interlaminar fracture toughness of CFRP laminates with GNP interleaves. The continuous GNP interlayer (with a 0.43 g/m$^2$ density) enhances the mode-II fracture toughness of CFRP by 40% but reduces mode-I toughness by $-31\%$. Improvement ($+42\%$) in interlaminar shear strength of the CFRP was found by Wang et al. [29] after introducing 10 wt% GNP/silicon carbide nanowire (SiCnw) interleaves. Korbelin et al. [30] investigated the dependence on the interlayer thickness of the interlaminar energy release rate under mode-I and mode-II loadings of few-layer graphene-modified CFRP with interlayer thicknesses varying from ultra-thin-ply (30 g/m$^2$) to thick-ply (240 g/m$^2$), showing a significant improvement of fracture toughness with respect to the neat CFRP in both cases.

In this study, a simple and scalable process based on spray deposition has been developed to functionalize the pre-impregnated carbon fibre layer, which can be inserted in the lamination sequence of composites without modifying their manufacturing process. The main advancement of this system is the possibility of adding a thick nanocomposite viscoelastic layer with improved damping capacity, by tuning both the areal weight and the filler/matrix composition. GNPs/epoxy paste was sprayed on the surface of the prepregs to fabricate a functionalized high-content GNP interlayer with a nominal content of 70 wt% of GNPs. Symmetric laminates have been fabricated by introducing the functionalized prepreg as a central ply in the stacking process. The effect of the GNP interlayer on the laminates' performances has been investigated by considering two different coating weights of 10 and 40 g/m$^2$. Both the viscoelastic behaviour of the CFRP laminates and the damage tolerance of the CFRP composite have been assessed by studying the effect of the GNP interleaves on the damping performance and mode-I and mode-II interlaminar fracture toughness and interlaminar shear strength (ILSS). A concurrent increase in damping capacity ($+25\%$) with an acceptable drop in fracture toughness ($-15\%$) at higher GNP content and compaction have been observed.

## 2. Materials and Methods

### 2.1. Integration of GNP-High-Content Layer onto Carbon Fibre Prepreg

GNPs/epoxy coating with a nominal weight ratio of 70/30 has been deposited on the surface of carbon fibre prepregs. Two-dimensional GNPs, with a high aspect ratio, called G2Nan (a lateral size of 30 μm and thickness of 14 nm), and the epoxy resin HexFlow® RTM6 were employed in this study. The carbon fibre/epoxy prepreg was purchased from Krempel. The 2 × 2 twill 3k carbon fibre/epoxy pre-impregnated layer (prepreg) is composed of a KGBD 2508 fibre (56% *w/w*) in an intermediate modulus epoxy with a 120 °C cure temperature and a $T_g$ (glass transition temperature) of 140 °C.

A spray deposition process is employed for coating the prepreg surface. Firstly, GNPs are dispersed in acetone by ultra-sonication and mixed with a solution of epoxy diluted in acetone previously prepared [15,31]. Then, obtained paste is deposited directly on the carbon fibre KGBD 2508 prepreg (30 × 30 cm) using a semiautomatic tri-axes pantograph, as shown in Figure 1. Finally, the material is dried at room temperature all night to let the solvent evaporate. Coating with two different areal weights, a low-weight GNP interlayer (LW-GNP) of 10 g/m² and high-weight GNP interlayer (HW-GNP) of 40 g/m², has been considered in this study (Figure 2). The areal weight of the coating has been managed by controlling the number of deposition cycles during the spraying process. Each deposition cycle corresponds to a nominal areal weight of 10 g/m².

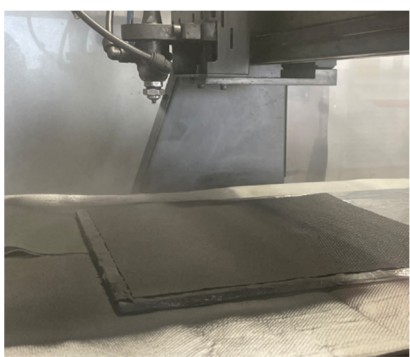

**Figure 1.** Spray deposition process of GNP coating on CFRP prepreg.

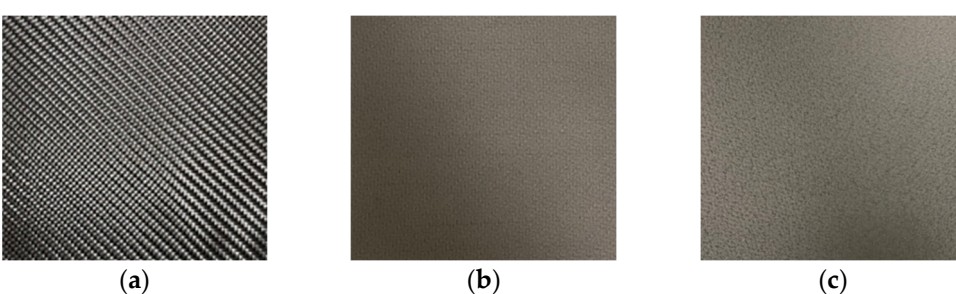

| (a) | (b) | (c) |

**Figure 2.** The surface of (**a**) prepreg (reference), (**b**) LW–GNP-coated prepreg, and (**c**) HW–GNP-coated prepreg.

### 2.2. Manufacturing of Hybrid Carbon Fibre Epoxy Composites

Large composite panels were prepared by stacking 16 plies of the KGBD 2508 prepreg with a $[(90/+45/-45/0)2]_S$ stacking sequence. The GNP-coated prepreg has been stacked at the eighth ply (mid-section), to ensure symmetry, as indicated in Figure 3a. Panels were cured in an autoclave under a vacuum on a flat mould (Figure 3b), with the additional pressure of 3 bar and heated at a rate of 3 °C/min to 100 °C for 120 min and subsequently post-cured at 140 °C for 60 min. Moreover, to create an initial crack for mode-I (ASTM D5528) and mode-II (ASTM D7905) fracture testing, a Teflon film of 13 μm thickness was placed between the 8th at the 9th ply across the top 65 mm of the sheet (Figure 3c). Panels

with LW-GNP and HW-GNP interlayers have been fabricated. A reference panel, without the GNP interlayer, has been fabricated for comparison. All fabricated samples are listed in Table 1.

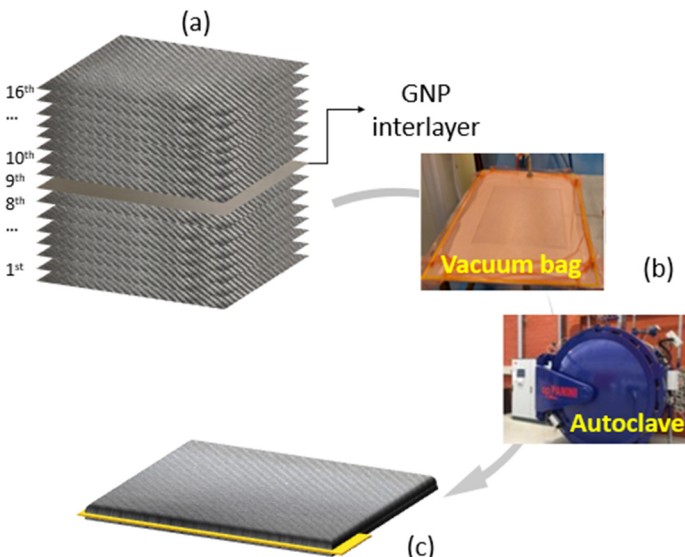

**Figure 3.** Lamination sequence (**a**); vacuum bagging and autoclave process (**b**); and CFRP panels with interleaved GNP layer (**c**).

**Table 1.** List of samples.

| Sample | Lamination Sequence | Filler/Matrix Content [*wt/wt*] | Number of Deposition Cycles | Coating Areal Weight [g/m$^2$] |
|---|---|---|---|---|
| REF | [(90/+45/−45/0)2]$_s$ | - | 0 | - |
| LW-GNP | [(90/+45/−45/0)2]$_s$ * | 80/20 | 1 | 10 |
| HW-GNP | [(90/+45/−45/0)2]$_s$ * | 80/20 | 4 | 40 |

* coated prepreg at the mid-section.

### 2.3. Experimental Characterization

A thermogravimetric analysis (TGA) (TA Instruments Q500) was conducted to evaluate the real filler/matrix composition of the coating. Measurements were performed in an inert atmosphere, using nitrogen gas, with a temperature ramp of 10 °C/min from room temperature to 800 °C, according to the ASTM E1131. The weight loss is evaluated at 600 °C.

The thermal properties of the material were investigated by differential scanning calorimetry (DSC) using the DSC Discovery instrument. Each specimen was heated and cooled twice from 0 to 300 °C at a rate of 10 °C/min.

A dynamic mechanical analysis (DMA) was employed to measure and assess the viscoelastic behaviour of the laminates. A TA Instruments Q800 DMA equipped with a 3-point bending clamp was used to perform a temperature sweep from 30 to 200 °C at a heating rate of 3 °C/min and a frequency of 1 Hz and considering an initial amplitude of 20 μm. Three tests were performed for each sample to ensure the repeatability of the result. Data are elaborated according to the ASTM D790 standard for the flexural behaviour of composites [32].

Finally, the composite specimens were cut by a diamond saw from the manufactured plates to perform ILSS and mode-I and mode-II fracture tests, using an Instron 68TM-50 universal testing apparatus.

The ILSS of composites was determined using the short-beam shear (SBS) test method following the ASTM D2344 standard. The interlaminar shear is generated indirectly through

the three-point bending of specimens in the SBS method. The tested specimens were compliant with the dimensions of $36 \times 12 \times h$ mm$^3$ where h was the actual thickness of the specimens. Three SBS samples were tested for each condition (REF, LW-GNP, and HW-GNP). The ILSS (MPa) values were calculated by Equation (1):

$$\text{ILSS} = \frac{3P_{max}}{4bh} \tag{1}$$

where $P_{max}$ (N) is the maximum load, $b$ (mm) is the sample width, and $h$ (mm) is the actual thickness of specimens.

For the mode-I fracture, composite specimens were prepared according to the ASTM D5528 std. with the dimensions $125 \times 25 \times 4.0$ mm$^3$, and an initial crack of 65 mm (Figure 4a). The edges of double cantilever beam (DCB) samples were painted with a white correctional fluid to improve crack visibility, and markings were added to track crack growth to the nearest millimetre. Steel loading blocks were glued to the ends of the sample beams using a cyanoacrylate adhesive. The bonding surface of the specimen has been lightly scrabbed with sandpaper and then wiped clean with methylethylketone (MEK) to remove any contamination.

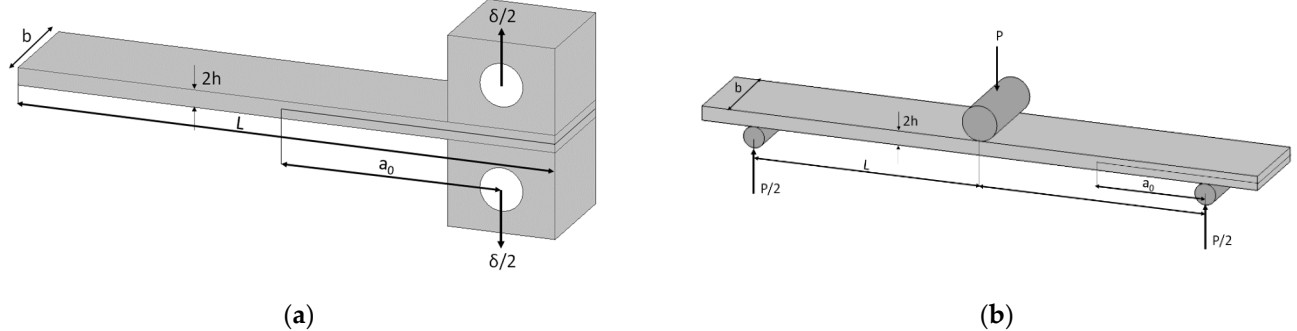

(**a**)                                                   (**b**)

**Figure 4.** Mode-I DCB (**a**) and mode-II ENF (**b**) testing configuration.

Three DCB samples were tested for each condition (REF, LW-GNP, and HW-GNP).

Mode-I fracture toughness, $G_{Ic}$, was calculated using the modified beam theory (MBT). MBT assumes a beam with no rotation at the reaction front. The test beam was loaded in the displacement control mode (2 mm/min), from the loading blocks until the crack front propagated at about 20 mm, before unloading. The beam theory expression for the strain energy release rate ($G_{Ic}$, kJ/m$^2$) of a DCB is as follows:

$$G_{Ic} = \frac{3P\delta}{2ba} \tag{2}$$

where $P$ (N) is the load, $\delta$ (mm) is the load point displacement, $b$ (mm) is the sample width, and $a$ (mm) is the crack length at the fracture.

The mode-II fracture was tested according to the ASTM D7905 std. on specimens with the dimensions of $160 \times 25 \times 4$ mm$^3$, and an initial crack of 45 mm. Three beams were tested for each condition. Mode-II interlaminar fracture toughness, $G_{IIc}$, was measured on a 3-point bending fixture as shown in Figure 4b. End-notch flexure (ENF) beams were set such that the crack tip was a fixed distance from one of the support rollers ($a_0 = 30$ mm) and loaded at 1 mm/min. Mode-II interlaminar fracture toughness, $G_{IIc}$ (kJ/m$^2$), is given by the following equation:

$$G_{IIc} = \frac{9a_0^2 P\delta}{2b\left(2L^3 + 3a_0{}^3\right)} \tag{3}$$

where $P$ (N) is the critical load, $a_0$ (mm) is the initial crack length, $\delta$ (mm) is the load point displacement, $L$ (mm) is the half span, and $b$ (mm) is the beam width.

## 3. Results and Discussion

### 3.1. Characterisation of Graphene-Coated Prepreg

The actual areal weight of the coating has been estimated to be 15 and 59 g/m$^2$ for LW-GNP and HW-GNP, respectively, as reported in Table 2. The actual value has been computed by measuring the difference in weight of the samples before and after the spray deposition process.

**Table 2.** DSC results on prepreg.

|  | Actual Areal Weight [g/m$^2$] | T$_{peak}$ [°C] | Peak Area [J/g] |
|---|---|---|---|
| REF | - | 145.13 | 394.80 |
| LW-GNP | 15 | 145.31 | 405.68 |
| HW-GNP | 59 | 148.70 | 371.43 |

Figure 5 shows the DSC curves of the analysis conducted on the prepreg samples and refers to the prepreg resin weight, which is the only reactive part of the system. In the pictures, a first peak located at 145 °C is found for all samples, which is associated with the resin of the Krempel prepreg. The peak area does not significantly modify in the case of a functionalized prepreg with respect to the reference. An increase of +3% was found in the case of LW-GNP and a decrease of −6% in the case of HW-GNP. This indicates that the spray deposition step does not affect the resin reactivity.

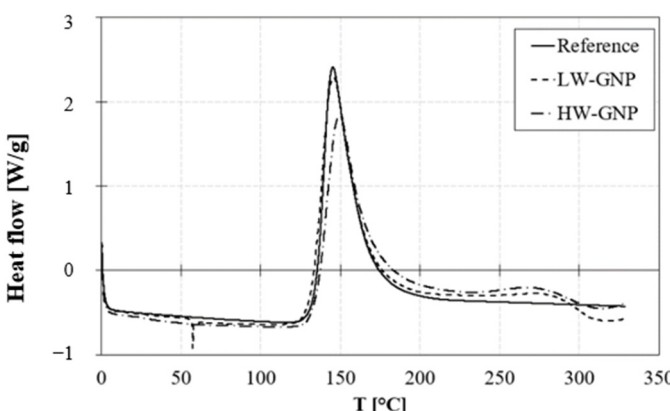

**Figure 5.** DSC curves in the heating phase (10 °C/min) for uncured prepreg samples.

Additionally, a second peak, located at 280 °C, is found in the case of LW-GNP and HW-GNP prepregs, which is associated with the RTM6 resin of the coating. The intensity of the peak is very low, due to the small amount of polymer present in the material.

### 3.2. Characterisation of Functionalized Laminates

#### 3.2.1. TGA, DSC, and DMA Results

Figure 6 shows the thermal degradation of laminates. A weight loss of 5% occurs at 370 °C for all samples and the weight residue at 600 °C is 50% for the reference sample and 60% for LW-GNP and HW-GNP samples.

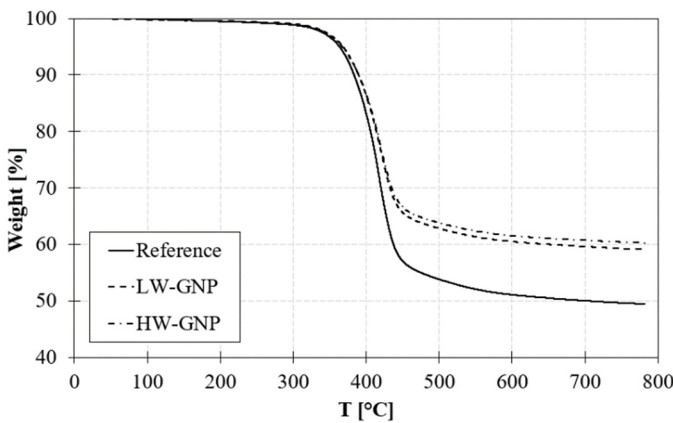

**Figure 6.** TGA curves of cured laminate samples.

DSC curves on cured laminates, referring to the Krempel resin weight (Figure 7), confirm the complete curing of the epoxy resin of Krempel, due to the absence of the reticulation peak at 145 °C. Nevertheless, the small peak at 280 °C, for LW-GNP and HW-GNP samples, indicates that the RTM6 resin is not completely cured.

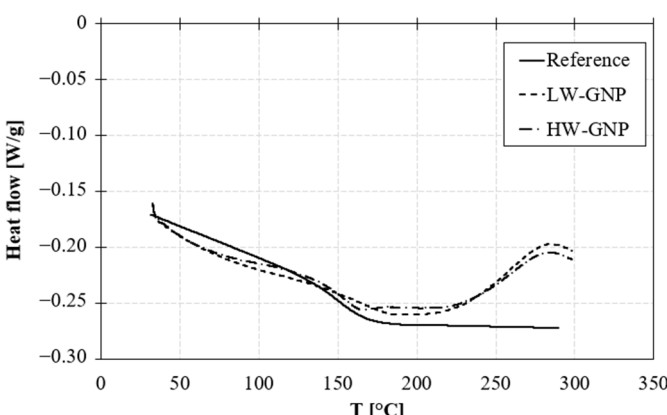

**Figure 7.** DSC curves in the heating phase (10 °C/min) for cured laminate samples.

The glass transition temperature ($T_g$) is not affected by the incorporation of GNPs. It slightly reduces by about 4% in the case of laminates with interleaved GNP layers regardless of the areal weight of the coating. The same trend is found for the $T_g$ obtained from the DMA analysis, as reported in Table 3.

**Table 3.** Results of test conducted on laminates.

|  | $T_{g,DSC}$ [°C] | $T_{g,DMA}$ [°C] | $E'$ [GPa] | $\Delta E'$ * [%] | $\tan\delta$ [-] | $\Delta\tan\delta$ * [%] |
|---|---|---|---|---|---|---|
| REF | 151.9 ± 0.5 | 159.4 ± 2.4 | 25.1 ± 1.6 | - | 0.024 ± 0.001 | - |
| LW-GNP | 145.1 ± 0.7 | 166.7 ± 0.8 | 23.2 ± 2.0 | −7 | 0.030 ± 0.007 | +25 |
| HW-GNP | 145.9 ± 0.2 | 164.4 ± 0.7 | 26.1 ± 2.5 | +3 | 0.025 ± 0.005 | +6 |

* $\Delta$ refers to the variation of the parameter with respect to the reference value.

The bending stiffness of the material increases when a thick interleaved layer is included in the laminate. In fact, the storage modulus ($E'$) slightly decreases (−7%) in the case of LW-GNP with respect to the reference and increases by +3% in the case of HW-GNP laminates at room temperature (Figure 8a).

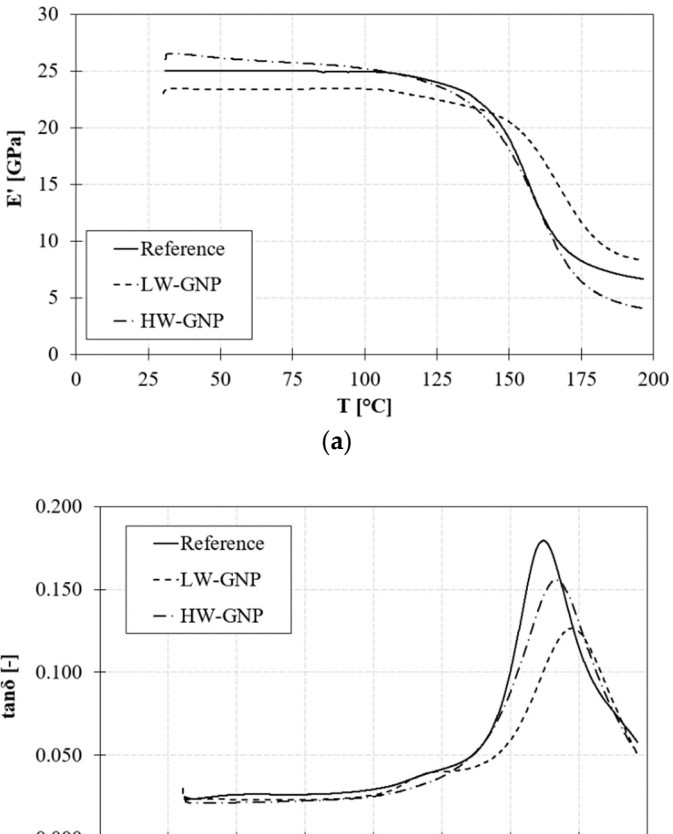

**Figure 8.** Storage modulus (**a**) and tanδ (**b**) of laminate samples.

The damping capacity of the material (tanδ) increases by +25% when a low-areal weight GNP interlayer is added. A lower increase in tanδ (+6%) is also shown for the HW-GNP sample (Figure 8b). On the contrary, at high temperatures, the peak damping parameter (tanδ peak) slightly decreases with the incorporation of GNPs in the interlaminar region, especially for the LW-GNP sample.

### 3.2.2. Interlaminar Properties

The results of interlaminar shear strength, and mode-I and mode-II fracture toughness of tested samples, are reported in Table 4.

**Table 4.** Results of ILLS, DCB, and ENF tests conducted on laminates.

|  | ILSS [MPa] | ΔILSS * [%] | $G_{IC, initial}$ [J/m$^2$] | $\Delta G_{IC, initial}$ * [%] | $G_{IC\ propagation}$ [J/m$^2$] | $\Delta G_{IC\ propagation}$ * [%] | $G_{IIC}$ [J/m$^2$] | $\Delta G_{IIC}$ * [%] |
|---|---|---|---|---|---|---|---|---|
| REF | 63.1 ± 0.3 | - | 174 ± 29 | - | 199 ± 12 | - | 1642 ± 206 | - |
| LW-GNP | 38.2 ± 1.4 | −40 | 92 ± 10 | −47 | 101 ± 8 | −49 | 532 ± 56 | −67 |
| HW-GNP | 34.5 ± 0.4 | −46 | 147 ± 29 | −15 | 128 ± 10 | −36 | 206 ± 5 | −87 |

* Δ refers to the variation of the parameter with respect to the reference value.

The ILSS reduces in the case of both LW-GNP and HW-GNP samples with respect to the reference. A higher reduction of −46% is found in the case of the high-areal weight GNP layer.

Figure 9 shows the relation between the applied load and displacement for REF laminates, LW-GNP, and HW-GNP. The curve modifies in the case of GNP interleaved samples with respect to the reference CFRP panel. Nevertheless, no significant differences are found between samples with a low- and high-areal weight interlayer. Initially, all the curves show an increase in load. After the initial increase in load, there is a large decrease marking the initiation of crack growth. This is again followed by a certain increase in load, only in the case of the reference, before it again starts decreasing. A noticeable difference in the crack propagation pattern is shown for LW-GNP and HW-GNP samples: the crack initiation happens at a higher displacement but at a much lower load level. With the addition of GNPs, the crack growth is much smoother than the reference sample, where many maxima–minims are shown, meaning that the crack propagation is unstable. This behaviour is due to a consistent resistance offered by cross-bridging between the GNPs and increases with increasing interlayer areal weight.

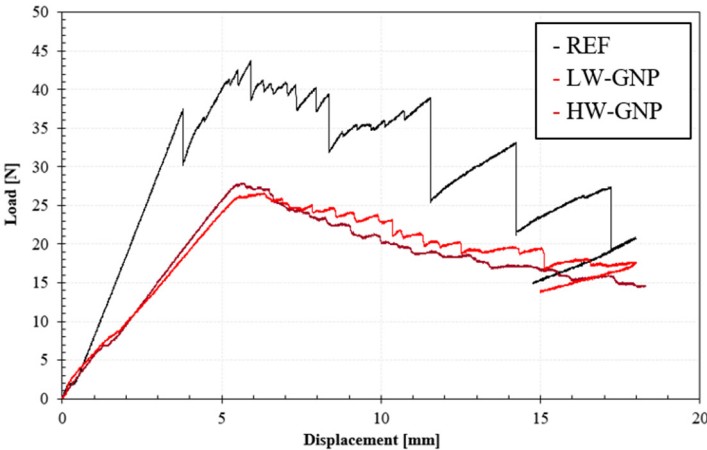

**Figure 9.** Load–displacement curves for representative fracture mode-I tests of reference CF/epoxy composites and LW-GNP and HW-GNP ones.

According to Equation (2), the mode-I fracture toughness ($G_{Ic}$) is calculated and the results are reported in Table 4. The relationship between mode-I fracture toughness ($G_{Ic}$) and the increment of the crack length is shown in Figure 10. Mode-I fracture toughness increases with the crack length as shown in Figure 10.

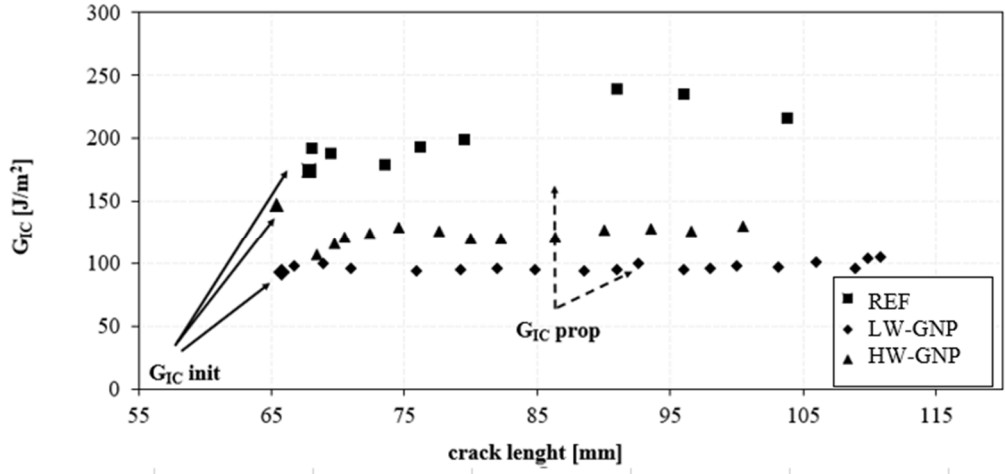

**Figure 10.** $G_{IC}$ values for the REF, LW-GNP, and HW-GNP specimens.

The first value of $G_{Ic}$ (on initiation) indicates the energy required for the initial crack extension and is defined as the point of the sudden decrease in load. The increase in

$G_{Ic}$ with the crack length is mainly due to fibre bridging that happens on the fracture surface and is indicated as $G_{Ic}$ on propagation (averaged over the first 25 mm of crack extension) [33].

The results confirm that the mode-I fracture toughness of the REF is higher than the samples with the GNP interlayer. The reduction in the mode-I toughness values can be attributed to the suppression of the fibre bridging. Additionally, the mode-I fracture toughness of the HW-GNP is higher than LW-GNP; these results confirm that the $G_{IC}$ is dependent on the area areal weight of the GNP interlayer.

Results for the end-notch flexural testing of the reference and GNP-treated CFRP are reported in Table 4. The mode-II fracture toughness is calculated according to Equation (3). A significant reduction is shown for LW-GNP and HW-GNP samples with respect to the reference.

### 3.3. Morphology of Fracture Propagation

Table 5 shows the micrographs of tested samples. Micrographs on ILSS-tested samples show that, unlike the reference sample, the fracture occurs within the GNP interlayer for both LW-GNP and HW-GNP, resulting in a cohesive fracture.

**Table 5.** Micrographs of tested samples (bulk and ILSS fractures).

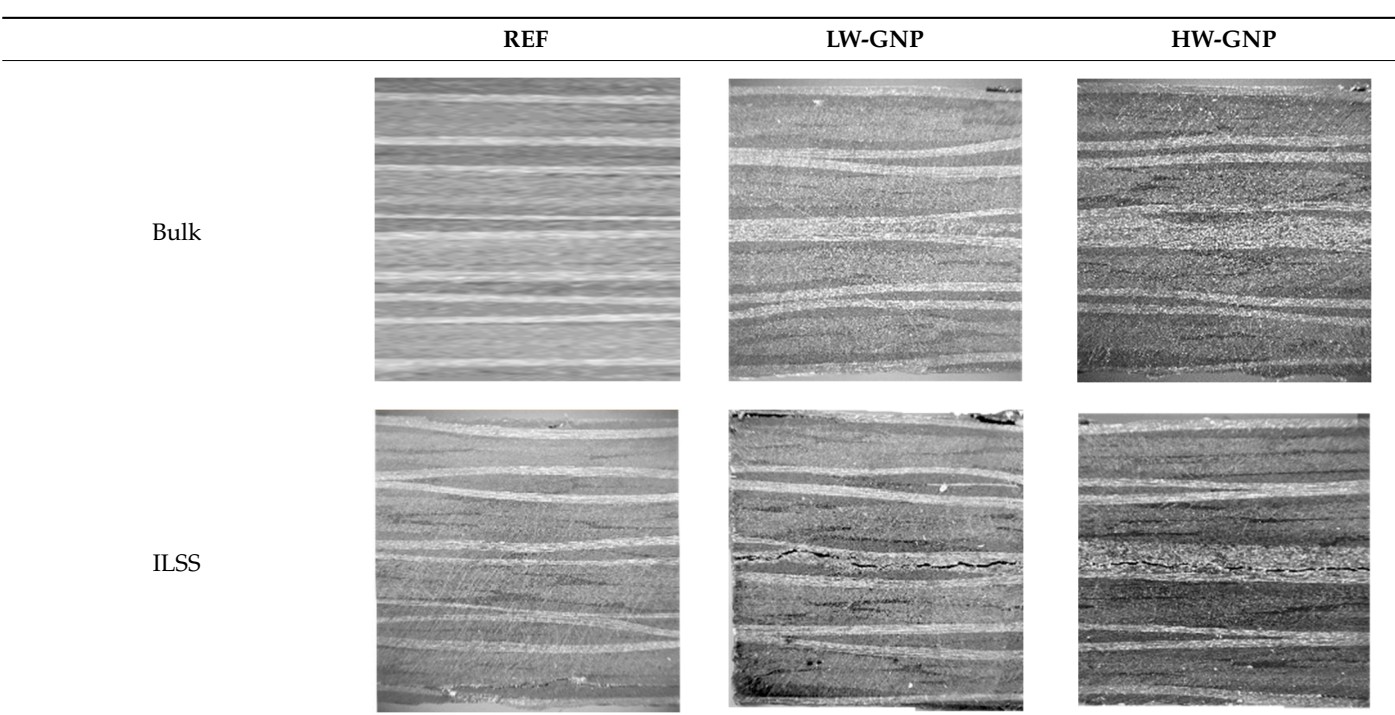

|  | REF | LW-GNP | HW-GNP |
|---|---|---|---|
| Bulk |  |  |  |
| ILSS |  |  |  |

From the ENF- and DCB-tested samples (Figures 11 and 12), both in the case of reference and LW-GNP samples, the fracture is cohesive. However, in the case of the HW-GNP sample, the fracture propagation occurs at the interface between the GNP layer and the adjacent CF/epoxy layer.

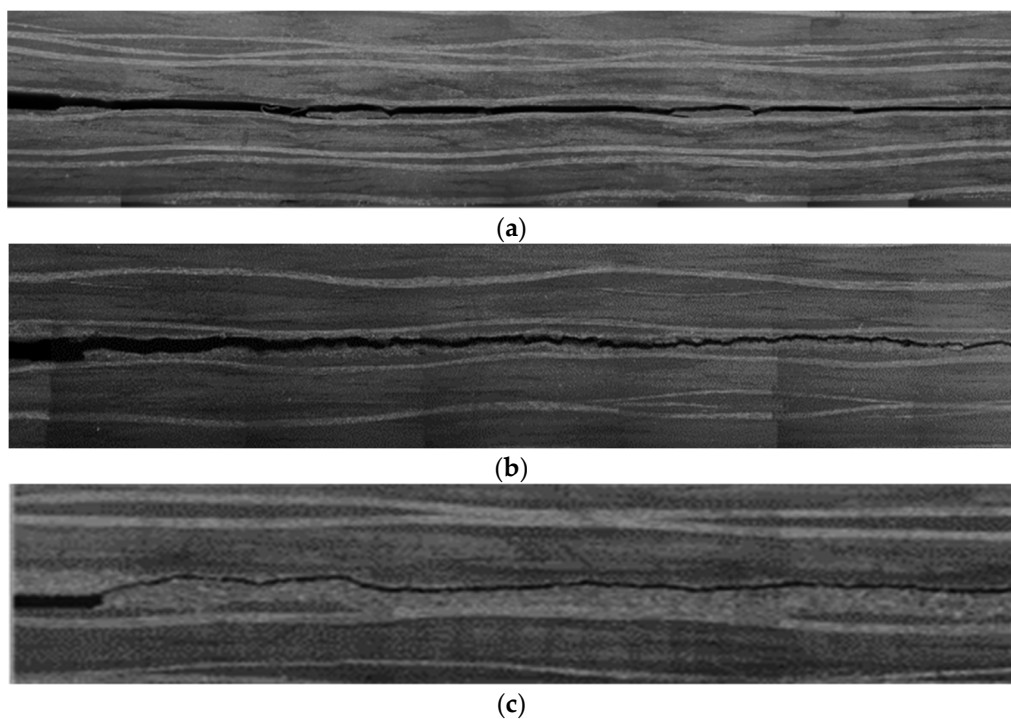

**Figure 11.** Micrographs of tested samples: ENF fractures of (**a**) REF; (**b**) LW-GNP; and (**c**) HW-GNP.

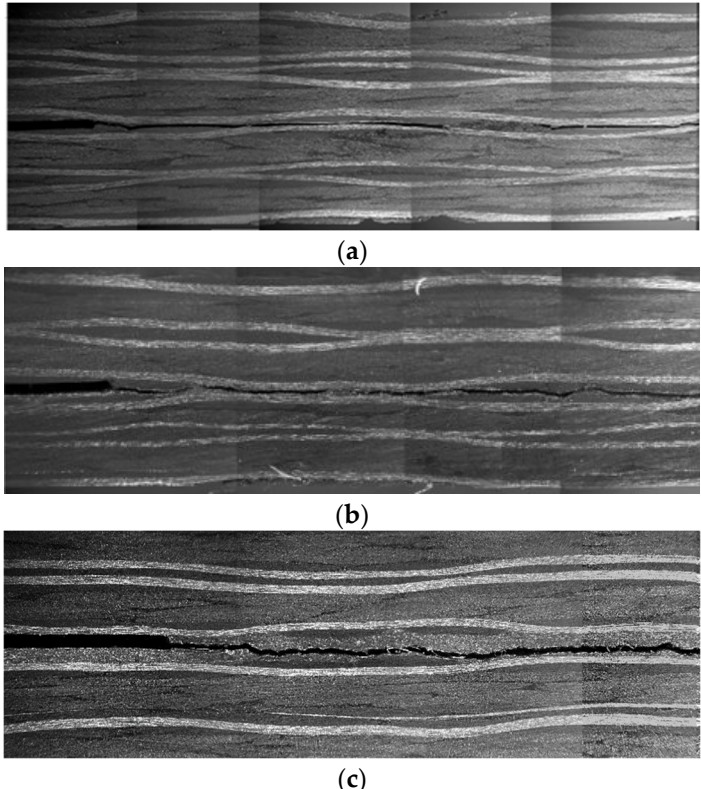

**Figure 12.** Micrographs of tested samples: DCB fractures of (**a**) REF; (**b**) LW-GNP; and (**c**) HW-GNP.

## 4. Discussion

The thickness of the GNP interlayer has been estimated by optical microscopy. The thickness of the GNP interlayer increases with the increasing coating areal weight, being 140 µm in the case of the LW-GNP sample and 320 µm in the case of HW-GNP; there is

no evidence of the diffusion of nanoparticles within the carbon fibre layers. Although the areal weight of the GNP coating is four times higher in HW-GNP compared to LW-GNP, the ratio between the two interlayer thicknesses is two, due to the effect of compaction pressure during the manufacturing of laminates (Figure 13).

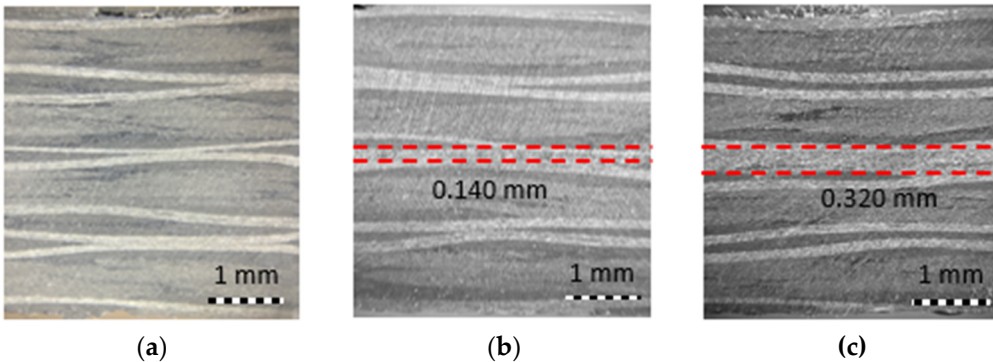

**Figure 13.** Specimen micrographs, and the GNP layer is highlighted by a dotted line. (**a**) REF—no GNP; (**b**) LW-GNP—140 microns; and (**c**) HW-GNP—320 microns.

Table 6 summarises the main properties (E′, tanδ, ILSS, $G_{CI}$, and $G_{CII}$) of all samples analysed at room temperature.

**Table 6.** Comparison of main properties of REF, LW-GNP, and HW-GNP samples.

| Property at RT | REF | LW-GNP | HW-GNP |
|---|---|---|---|
| E′ [GPa] | 25.1 ± 1.6 | 23.2 ± 2.0 | 26.1 ± 2.5 |
| tanδ [-] | 0.024 ± 0.001 | 0.030 ± 0.007 | 0.025 ± 0.005 |
| ILSS [MPa] | 63.1 ± 0.3 | 38.2 ± 1.4 | 34.5 ± 0.4 |
| $G_{CI}$ [J/m$^2$] | 174 ± 29 | 92 ± 10 | 147 ± 29 |
| $G_{CII}$ [J/m$^2$] | 1642 ± 206 | 532 ± 56 | 206 ± 5 |

From the results of the DMA analysis, it is found that the elastic behaviour of CF/epoxy laminates is not significantly affected by the high-content-GNP interlayer since the storage modulus varies in a small range (−7%, +3%). On the contrary, the dissipative behaviour of the laminates increases by +25% in the case of LW-GNP and +6% in the case of HW-GNP interlayers. The GNP interlayer acts as a soft layer, improving the dissipation of vibrational energy of the laminates [4]. Thanks to the high gradient stiffness (from the CF/epoxy to GNP/epoxy layer), greater interlaminar stresses are concentrated in the GNP layer, which dissipates energy through interlaminar damping [34].

However, the interlaminar fracture of both the low- and high-areal weight GNP interlayer decreases with respect to the CF/epoxy laminate. In ILSS, the tensional state is mainly governed by the transverse shear load. Results demonstrate a worsening effect of the nanofiller as the areal weight increases due to the fact that the shear stresses, proportional to the section area, decrease as the thickness of the interlayer increases (Table 5).

By increasing the areal weight, and subsequently increasing the GNP layer thickness, the GNP layer acts as a barrier film since the tortuosity is highly increased, limiting the resin to flow toward the next layer [31,35]. The fracture progression confirms the lack of resin flow across the GNP layer (Figures 11 and 12); indeed, the failure occurs inside the GNP interlayer, and it is confirmed by the dry state of the fractured surfaces. Similar results were found by Moustapha Sarr et al. [36] in cellulose nanofibre (CNF) in glass fibre/epoxy composites, where the incomplete impregnation of glass fibre at the interface occurred when a thick CNF layer was added. This results in low interfacial adhesion.

Similar effects are found on the fracture toughness of the composites. The mechanism through which interlayered GNPs can affect composites' mode-I fracture toughness is

dual, both improving the fracture toughness of resin and acting as crack bridging [26]. As the crack propagates through the composite, the GNP interlayer contributes to uniformly distributing the load, as shown in the load–displacement curve, where a smoother and more stable trend is reported for LW-GNP and HW-GNP samples compared to the reference. The high surface area of the GNPs can promote the creation of a large number of microcracks, which can help to absorb energy and prevent uncontrollable failure of the composite material. In well-aligned 2D nanoplatelet composites, failures should occur at different dimensional scales: at the interface with the carbon fibre, within the GNP layer separating nanoparticles, and failures within GNP particles [37]. However, the mode-I fracture toughness reduces by −40% and −15% in the case of LW-GNP and HW-GNP samples, respectively, compared to the reference. This reduction may be due to the orientation of the nanoplatelets in the fibre-reinforced specimens. The spray deposition process of high-loaded GNP coating promotes the nanoplatelets' alignment on the surface of the prepreg. In addition, the confinement induced by the fibre layers (before the resin cure), due to the vacuum and external pressure applied on the mould during fabrication, contributes to the nanoplatelets' alignment, especially near the interfaces [38]. The higher the achieved alignment during deposition, the lower the capability of the nanoparticle to diffuse inside the CFRP layers, preventing the GNPs from bridging. In fact, in the case of the low-areal weight interlayer (Figure 12), the fracture toughness is weak. In the LW-GNP sample, the crack propagates in the planar direction of the laminate through the GNP layer. In contrast, in the HW-GNP sample, the crack initially propagates in the middle section of the GNP layer and then deviates to the deposition interface. The different behaviour observed for low- and high-areal weight is clearly associated with the nanoplatelet's alignment. The preparation of LW-GNP deposition on CFRP facilitates a well-aligned assembly of GNP, resulting in poor fracture toughness. The preparation of HW-GNPs requires multiple spray stages, which promotes the rise of misalignment on the GNP stacking; therefore, the nanoplatelets' alignment is reduced, and a random orientation is favoured within the GNP layer. In this case, a crack bridging effect is barely activated, leading to a lower decrease in $G_{IC}$ (−15%).

The fracture surfaces confirm the role of the GNP dispersion state on the final performances. Figure 14 shows for each sample the dispersion state, and the right side is the layer where the GNPs have been deposited. In the case of LW, the GNP stacking is well-ordered and the effect of the compression between layers (upper and lower) is clearly visible. In the case of HW, the GNP layer loses its alignment. It is worth noting that in both cases, the fracture propagates inside the GNP layer (Figure 12).

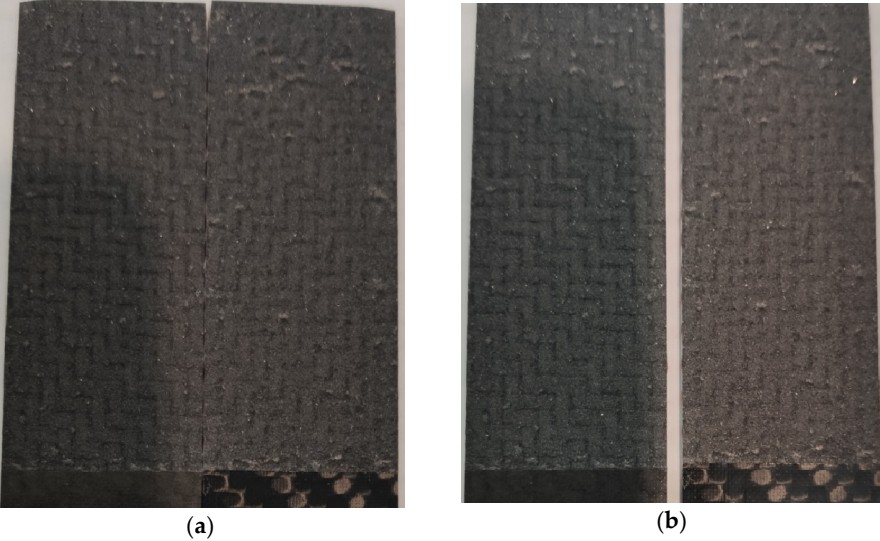

(a)          (b)

**Figure 14.** Fracture surfaces for mode-I specimens. (**a**) LW-GNP and (**b**) HW-GNP layer.

Similarly, ENF results showed a significant reduction in mode-II fracture toughness. The evaluation of fracture and toughening mechanisms under the mode-II fracture is relatively more challenging when compared to the mode-I fracture. When a crack, originally in the mode-II state, starts propagating, it often changes to the opening mode (i.e., the mode-I fracture) [38]. Therefore, shortly after the crack propagation starts, the observed fracture surface patterns become very similar to those observed for the mode-I fracture.

The crack propagation path (Figure 11) in mode-II loading shows a global behaviour similar to the double cantilever, even if the presence of a thick layer in the case of HW samples gives rise to an initial failure path related to intensification around the initial delamination corner, and subsequently propagates along the GNP layer thickness, suggesting that the tensional state is more similar to a mixed-mode fracture mechanism rather than mode-II [38].

## 5. Conclusions

In this work, the effect of a GNP interlayer on the mechanical and fracture behaviour of CFRP laminates has been investigated. The influence of layer thickness (i.e., areal weight) has been studied. The GNP interlayer offers improvement in the dissipation mechanism, without affecting the elastic modulus of the laminate, thanks to the intrinsic damping capacity of high-aspect ratio GNPs and the high gradient stiffness between the CF/epoxy layer and the GNP/epoxy layer. The LW-GNP layer led to an increase in the damping capacity of 25%, while HW-GNP has a damping capacity slightly higher than the reference system (+6%). However, the presence of interleaves does not improve the interlaminar fracture toughness of laminates. The mode-I fracture tends to take place cohesively through the GNP layer in the case of the LW-GNP sample and at the interface with the deposited GNP layer in the case of the HW-GNP sample. In both cases, an abrupt decrease in the critical energy release rate was experienced, $-49\%$ and $-15\%$, respectively. The mode-II fracture follows the same propagation crack as mode-I since $G_{IC}/G_{IIC} < 1$. Even the mode-II performances were negatively affected by the presence of the GNP layer with a reduction of 67% and 87% in the critical energy.

The obtained results suggest that spray deposition technology is suitable to realize the functional layer (epoxy/CF prepregs modified with GNP layer depositions) at different areal weights, improving the damping of CF/epoxy composites; however, fracture mechanics require further investigation to preserve the initial performances.

Future works should aim to simultaneously improve the damping and fracture toughness by modifying the areal weight and/or the GNP aspect ratio. Furthermore, the effect of the interlayer on thermal and electrical conductivities can be investigated.

**Author Contributions:** Conceptualization, A.M. and F.C.; methodology, B.P. and A.M.; investigation, F.C., B.P. and C.S.; data curation, A.P. and M.E.; writing—original draft preparation, F.C. and B.P.; writing—review and editing, A.M. and M.G.; supervision, A.M.; funding acquisition, A.P. and M.G. All authors have read and agreed to the published version of the manuscript.

**Funding:** This research was carried out in the framework of the project MUSAICO, grant number F/190014/01-02/X44, founded by the Italian Government Data.

**Data Availability Statement:** Not applicable.

**Acknowledgments:** The authors would like to thank ATM srl for supporting manufacturing of CFRP in an industrial relevant environment and Jaber Innovation srl for supporting the setup of GNP interlayer deposition. The authors would like to thank Giuseppe De Tommaso and Francesco Bertocchi for their valuable advice and comments on various technical issues.

**Conflicts of Interest:** The authors declare no conflict of interest.

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
