# Peer review of "Mechanical and Viscoelastic Properties of Carbon Fibre Epoxy Composites with Interleaved Graphite Nanoplatelet Layer"

_jcs, doi:10.3390/jcs7060235_

Round 1

Reviewer 1 Report

It is a useful but dull and lengthy manuscript. I would recommend to separate the main results and the rest put into the Supplement Section.

I have only minor comments on the presentation.

line 13. The reader needs to guess what is CFRP

lines 28-107. The Introduction section is absolutely unacceptable of such length. It is a review of the literature. The authors should briefly formulate the problem, which will be tackled below.

line 18. The authors confuse weight (newtons, N) and mass (kg). Weight is force.

lines 17, 132, others. Prepreg. The general reader does not know what it is.

line 114. 3k prepreg ?

line 312. barbaric optical spectroscopy ?

There should be a space between the value of temperature and the sign of temperature (o) (IUPAC).

Need an additional minor revision.

Author Response

It is a useful but dull and lengthy manuscript. I would recommend to separate the main results and the rest put into the Supplement Section.

The authors are grateful to the reviewer for the comments. The authors have been able to incorporate changes to reflect most of the suggestions provided.

I have only minor comments on the presentation_

  1. line 13. The reader needs to guess what is CFRP.

fixed

  1. lines 28-107. The Introduction section is absolutely unacceptable of such length. It is a review of literature. The authors should briefly formulate the problem, which will be tackled below.

The introduction section has been shortened to outline the main questions investigated within the paper.

  1. line 18. The authors confuse weight (newtons, N) and mass (kg). Weight is force.

The authors adopted a commonly used definition for fiber reinforced composites. The Areal weight refers to the mass/weight of reinforcement per unit area. (Typically, in g/m2 gsm) or ounces/yard2. The abstract was revised to avoid misunderstanding.

  1. lines 17, 132, others. Prepreg. The general reader does not know what it is.

The abstract was revised to avoid misunderstanding.

  1. line 114. 3k prepreg ?

The “3k” refers to the woven carbon fiber reinforcement and indicates that each carbon fiber tow includes 3000 single carbon fiber filaments.

  1. line 312. barbaric optical spectroscopy ?

It was a typo, fixed.

  1. There should be a space between the value of temperature and the sign of temperature (o) (IUPAC).

fixed

Reviewer 2 Report

The article presents very interesting experimental results, and can be published almost as sent, after addressing only a few minor comments:
1. please clearly emphasize in the abstract or introduction the novelty of the present work in relation to other thematically similar research papers.
2. please expand the literature review in the introduction to include papers from the described research topic, among others (DOI): 10.12913/22998624/67677, 10.12913/22998624/64006.
3 Figure 3 is presented in an unrepresentative manner. Please both enlarge, and take care and better quality of presentation.
4. In conclusion, please include conclusions relating not only to qualitative evaluation (what was done) but also quantitative.

Author Response

The article presents very interesting experimental results, and can be published almost as sent, after addressing only a few minor comments:

Thank you for your comment, the authors have incorporated changes to address your suggestions.

  1. please clearly emphasize in the abstract or introduction the novelty of the present work in relation to other thematically similar research papers.

The abstract and the introduction section have been revised to outline the main questions investigated and clearly present the progress with respect to the state-of-the-art.

  1. Please expand the literature review in the introduction to include papers from the described research topic, among others (DOI): 10.12913/22998624/67677, 10.12913/22998624/64006.

The introduction section has been improved by including relevant references.

  1. Figure 3 is presented in an unrepresentative manner. Please enlarge both and take care and better quality of presentation.

The picture has been edited to improve its quality.

  1. In conclusion, please include conclusions relating not only to qualitative evaluation (what was done) but also quantitative.

The conclusion paragraph has been improved by including quantitative evaluation of results achieved.

Reviewer 3 Report

It is an original paper dealing with “Mechanical and Viscoelastic Properties of Carbon Fiber Epoxy Composites with Interleaved Graphite Nanoplatelet Layer “. Regarding this manuscript there are some minor and major comments below to help the readers to be more beneficial from the paper.

1.      In the abstract, provide a summary statement that highlights the significance and implications of the objectives and findings.

2.      In introduction line 58, alongside ref [16], for further information refer to the reference below

[a] Ghabezi, P., Farahani, M., & Hosseini Fakhr, M. (2016). Investigation of mechanical behavior of alfa and gamma nano-alumina/epoxy composite made by vartm. Int J Adv Biotechnol Res7, 731-736.

3.      The introduction, line 62, 63 and 64, the authors spoke about the exceptional properties of GNP for reinforcing of composite material without addressing. It is necessary to address this sentence to the references below.

[b] Namdev, A., Telang, A., & Purohit, R. (2022). Experimental investigation on mechanical and wear properties of GNP/Carbon fiber/epoxy hybrid composites. Materials Research Express9(2), 025303.

[c] Sam-Daliri, O., Farahani, M., Faller, L. M., & Zangl, H. (2020). Structural health monitoring of defective single lap adhesive joints using graphene nanoplatelets. Journal of Manufacturing Processes55, 119-130.

[d] Bordoloi, M. M., Kirtania, S., Kashyap, S., & Banerjee, S. (2022). Investigation of Mechanical Properties of Carbon Fiber/Graphene Nanoplatelet/Epoxy Hybrid Nanocomposites. In Recent Advances in Materials Processing and Characterization: Select Proceedings of ICMPC 2021 (pp. 211-223). Singapore: Springer Nature Singapore.

4.      In line 138, write the standards of fracture testing (Mode and II) by addressing suitable reference

5.      Line 171, equation number is missed.

6.      In line 254, the authors say “With increasing coating areal weight, the GNPs layer becomes thicker, preventing the epoxy resin from impregnating the CF fibres at the interface.” Is there any evidences about this form your result?

7.      The authors brought some scientific information in the manuscript but there was no evidence for that or it’s not indicated by references for example:

“The increase of GIc with crack length is mainly due to fibre bridging that happens on the fracture surface and is 281 indicated as GIc on propagation (averaged over the first 25 mm of crack extension)”, where fibre bridging is shown in the manuscript. In the pictures Fibre Bridge is not indicated.

Author Response

It is an original paper dealing with “Mechanical and Viscoelastic Properties of Carbon Fiber Epoxy Composites with Interleaved Graphite Nanoplatelet Layer “. Regarding this manuscript there are some minor and major comments below to help the readers to be more beneficial from the paper.

The authors are grateful to the reviewer for insightful comments. The authors have been able to incorporate changes to reflect most of the suggestions provided.

  1. In the abstract, provide a summary statement that highlights the significance and implications of the objectives and findings.

The abstract has been revised to outline the main questions and achievements.

  1. In introduction line 58, alongside ref [16], for further information refer to the reference below
    [a] Ghabezi, P., Farahani, M., & Hosseini Fakhr, M. (2016). Investigation of mechanical behavior of alfa and gamma nano-alumina/epoxy composite made by vartm. Int J Adv Biotechnol Res, 7, 731-736.

            The introduction section has been improved by including relevant references.

  1. The introduction, line 62, 63 and 64, the authors spoke about the exceptional properties of GNP for reinforcing of composite material without addressing. It is necessary to address this sentence to the references below.

[b] Namdev, A., Telang, A., & Purohit, R. (2022). Experimental investigation on mechanical and wear properties of GNP/Carbon fiber/epoxy hybrid composites. Materials Research Express, 9(2), 025303.

[c] Sam-Daliri, O., Farahani, M., Faller, L. M., & Zangl, H. (2020). Structural health monitoring of defective single lap adhesive joints using graphene nanoplatelets. Journal of Manufacturing Processes, 55, 119-130.

[d] Bordoloi, M. M., Kirtania, S., Kashyap, S., & Banerjee, S. (2022). Investigation of Mechanical Properties of Carbon Fiber/Graphene Nanoplatelet/Epoxy Hybrid Nanocomposites. In Recent Advances in Materials Processing and Characterization: Select Proceedings of ICMPC 2021 (pp. 211-223). Singapore: Springer Nature Singapore.

The introduction section has been improved by including relevant references.

  1. In line 138, write the standards of fracture testing (Mode and II) by addressing suitable reference.

The statement has been modified by introducing the standard method and its reference.

  1. Line 171, equation number is missed.        
    fixed
  2. In line 254, the authors say “With increasing coating areal weight, the GNPs layer becomes thicker, preventing the epoxy resin from impregnating the CF fibres at the interface.” Is there any evidences about this form your result?

The statement has been reformulated to clarify. The manuscript was modified as follow:

“By increasing the areal weight, and the subsequent increase of the GNP layer thickness the GNP layer acts as a barrier film since the tortuosity is highly increased limiting the resin to flow toward the next layer [31,33]. The fracture progression confirms the lack of resin flow across the GNP layer (Figure 11 and Figure 12) indeed the failure occurs inside the GNP interlayer, and it is confirmed by the dry state of the fractured surfaces. Similar results were found by Moustapha Sarr et al. [34] in cellulose nanofiber (CNF) in glass fibre/epoxy composites, where the incomplete impregnation of glass fibre at the interface occurred when a thick CNF layer was added.”

  • Cilento, F.; Curcio, C.; Martone, A.; Liseno, A.; Capozzoli, A.; Giordano, M. Effect of Graphite Nanoplatelets Content and Distribution on the Electromagnetic Shielding Attenuation Mechanisms in 2D Nanocomposites. J. Compos. Sci. 2022, 6, 257;
  • Cui, S.I. Kundalwal, S. Kumar, Gas barrier performance of graphene/polymer nanocomposites, Carbon N. Y. 98 (2016) 313–333. https://doi.org/10.1016/j.carbon.2015.11.018
  • Moustapha Sarr, M.; Kosaka, T. Effect of cellulose nanofibers on the fracture toughness mode II of glass fiber/epoxy composite laminates. Heliyon 2023, 9, e13203, doi:10.1016/j.heliyon.2023.e13203.  
  1. The authors brought some scientific information in the manuscript but there was no evidence for that or it’s not indicated by references for example:

“The increase of GIc with crack length is mainly due to fibre bridging that happens on the fracture surface and is 281 indicated as GIc on propagation (averaged over the first 25 mm of crack extension)”, where fibre bridging is shown in the manuscript. In the pictures Fibre Bridge is not indicated.

A reference has been added to the manuscript to strengthen the concept. Even if, In the present case the bridging effect is not relevant as stated in the discussion section (row 374):  “…In this case, a crack bridging effect is barely activated leading to a lower decrease of GIC (-15%) is found.”

Round 2

Reviewer 1 Report

They should explain what CFRP is, line 13. Otherwise, everything is OK with me.

They should explain what CFRP is, line 13. Otherwise, everything is OK with me.

Author Response

Thank you for your comment, the authors have incorporated required changes. 

Reviewer 2 Report

The paper may be accepted.

Author Response

Thank you for your comments. The manuscript was improved by addressing your suggestions.

Reviewer 3 Report

Accept in present  form

Author Response

Thank you for your comments. the manuscript was improved by addressing your suggestions.